# Animal Personality and Conservation: Basics for Inspiring New Research

**DOI:** 10.3390/ani11041019

**Published:** 2021-04-04

**Authors:** Cristiano Schetini de Azevedo, Robert John Young

**Affiliations:** 1Departamento de Biodiversidade, Evolução e Meio Ambiente, Instituto de Ciências Exatas e Biológicas, Campus Morro do Cruzeiro, Universidade Federal de Ouro Preto, s/n Bauxita, Ouro Preto, MG 35.400-000, Brazil; 2School of Science, Engineering and Environment, University of Salford Manchester, Peel Building—Room G51, Salford M5 4WT, UK; r.j.young@salford.ac.uk

**Keywords:** boldness, copying style, extinction, temperament, translocation

## Abstract

**Simple Summary:**

The study of animal personality is important to conserve animals because it can help in selecting the most appropriate individuals to be released into the wild. Individuals not so bold or aggressive, less stressed, who explore their new environment with greater caution are often more likely to survive after release into the wild. In contrast, bolder and more aggressive animals reproduce more successfully and, therefore, can be released with the aim of rapid repopulation of an area. These and other aspects of how animal personality can help in conservation programs, as well as how to collect personality data are covered in this paper.

**Abstract:**

The number of animal species threatened with extinction are increasing every year, and biologists are conducting animal translocations, as one strategy, to try to mitigate this situation. Furthermore, researchers are evaluating methods to increase translocation success, and one area that shows promise is the study of animal personality. Animal personality can be defined as behavioral and physiological differences between individuals of the same species, which are stable in time and across different contexts. In the present paper, we discuss how animal personality can increase the success of translocation, as well as in the management of animals intended for translocation by evaluating personality characteristics of the individuals. Studies of the influence of birthplace, parental behavior, stress resilience, and risk assessment can be important to select the most appropriate individuals to be released. Finally, we explain the two methods used to gather personality data.

## 1. Introduction

The increase in the number of animals threatened by extinction is leading researchers worldwide to increase their efforts to conserve animal species [1]. Among the conservation initiatives, habitat protection and captive breeding for animal translocation/reintroduction are the most applicable since they have direct implications for the survival of the species in a specific location [2,3,4,5]. Animal introduction (i.e., the intentional movement and release of an organism outside its indigenous range), reintroduction (i.e., the intentional movement and release of an organism inside its indigenous range from which it has disappeared), and reinforcement (i.e., the intentional movement and release of an organism into an existing population of conspecifics) are types of animal translocation (all definitions taken from [6]).

The success of animal translocations can be low [7,8,9] and biologists are studying methods to decrease failures in animal translocation programs [10,11,12]. One of the actions is to consider how aspects of the animals’ behavior affects conservation success [13,14,15]. Conservation biologists have been working separated from behavioral biologists for a long time and the recent union of both scientific areas has proved to be important for the increase in the success rates of translocation programs [16,17,18]. A translocation is considered a success if the released animals maintain a viable population in the release area [6].

Animal releases can occur by two methods: soft or hard release [19,20]. There are two main differences between them: (1) pre-release management, which is implemented during soft release and is not implemented during hard release [19,21,22], and (2) post-release support, which is also implemented after soft release and comprises food and shelter provisioning/supplementation [6,23,24]. The lack of implementation of pre-release management and post-release support for hard release is because hard release generally is used when wild-caught animals are released or when large numbers are released, as in the case of fish, amphibians, etc. [19,25]. Most of the animals receiving pre-release management are raised or have spent most of their lives in captivity [19,26,27,28].

Populations of captive-born animals are being released in translocation programs [29,30,31], but problems linked to life in captivity are crucial to the failure of the programs [32,33]. For example, captive-born animals may lack the behavioral skills needed to survive after release, such as the ability to find food, shelter, or mates [34,35]. Sometimes captive-born animals cannot identify and respond to their predators [36,37]. Also, behavioral problems, such as the performance of abnormal behaviors may compromise individuals’ survival after translocation [38,39,40]. Finally, artificial/unconscious selection of certain personality profiles in captivity can affect the evolutionary potential of reintroduced populations [41].

Behavioral problems can be managed before the release of the animals into nature, for example, by animal training or environmental enrichment [41,42]. Furthermore, the use of tools that allows the choice of the most suitable individuals for release should greatly improve translocation success. Pre-release anti-predator conditioning [43,44,45], appropriate environmental enrichment [46,47], food and movement training [24,48] are among pre-release management techniques, which enhance the animals’ skills. Investigating aspects of the animals’ personalities is one of the tools that is also being applied to try and identify the most appropriate individuals that should be release into nature in terms of conservation success [49,50,51].

Animal personality can be defined as behavioral and physiological differences between individuals of the same species that are stable in time and across different contexts [52,53,54]. Animal temperament, coping styles, behavioral syndromes, and behavioral predisposition are synonyms of animal personality found in the scientific literature [52,53,54]. Every person who has two or more vertebrate pets can easily distinguish differences in their personalities, and these differences are beginning to be observed and studied for wild animals, both in nature and in captivity [54]. Personality is a construct, identified and classified for animals as including: neuroticism, agreeableness, extraversion, openness, conscientiousness, dominance, boldness, sociability, activity, exploration, aggressiveness, and activity, with some overlaps [52,55,56]. Although linking animal personality and conservation is relatively new for science [54], it has proven to be promising. In this paper, we will discuss how the study of animal personality can be important for animal conservation.

## 2. Animal Personality and Animal Releases

As mentioned, previously, animal translocation is one of the conservation measures used to avoid animal extinctions. Also, the use of captive-born animals is frequent because, in addition to providing individuals for release, it is in agreement with the conservation goals of modern zoos [57,58,59]. Examples of successful translocation programs that released captive-born animals are: the golden-lion tamarin *Leontopithecus rosalia* [60]; the black-footed ferret *Mustela nigripes* [61]; the Arabian oryx *Oryx leucoryx* [62]; and the bald eagle *Haliaeetus leucocephalus* [63]. However, a bias towards reporting successful translocation programs is known in the scientific literature [64,65], which means that we do not know the number of translocation programs that fail. Furthermore, many more translocation programs are undertaken without scientific evaluation, as a legal requirement, in response to anthropogenic activities such as dam building [20].

Within this scenario, would it be possible to use personality to enhance translocation success? The interest in linking personality directly to animal translocation emerged after a study of reintroduced swift foxes (*Vulpes velox*) in Canada found that survival was linked to personality [66]. Six weeks after release, the monitoring of 31 reintroduced swift foxes showed that bolder individuals died sooner than shyer individuals. This study was a pivotal moment for conservation efforts because it shows that shyer individuals were best to be translocated first, since they have a greater probability of surviving longer and, consequently, establishing in the area. However, why did this happen and is this true for all species? After the publication of this study, there have been calls in the literature to include personality assessment in translocations/animal releases, and boldness, exploration, activity, and sociability became the most studied aspects of personality-conservation dyad [41,67,68,69].

### 2.1. Boldness, Exploration, and Activity: Influence on Survival after Release

Bolder animals are more prone to take risks, because they explore more their environment, get closer to predators, sample more food items, etc. [66,68,70]. By doing this, bolder animals may die sooner than shyer animals [66]. However, studies show they reproduce more when compared to shyer individuals [71,72]. Furthermore, exploration of the environment can be advantageous for the individual to build a mental map of the area, locating feeding areas, shelter areas, mates, etc. [73,74,75]. Thus, there are advantages and disadvantages of reintroducing only shyer individuals.

In a study of 23 captive-born Blanding’s turtles (*Emydoidea blandingii*), researchers found that turtles presented different personality traits in the aggressiveness, boldness, and exploratory dimensions [68]. After reintroduction, more exploratory turtles survived more than less exploratory turtles, the exploratory dimension being the only one that influenced survival. Exploratory turtles used significantly more muskrat dens to hide and rest, and this could be responsible for their greater survival after release.

In a study with 28 Tasmanian devils (*Sarcophilus harrissii*) in Australia, bold individuals survived 3.5 times longer after translocation than shy individuals [76]. The researchers suggest that this result could be due to environmental spatial and temporal conditions at the time of release since fitness landscapes of behavior changes over time [77,78,79,80]. In other words, it is possible that the relationship between boldness and survival could change in different temporal and spatial circumstances; that is, shyer individuals could be favored in different years or/and in different locations.

Other studies evaluated the influence of personality on the survival of the released animals. In a study of blue-fronted parrots (*Amazona aestiva*), shyer and bolder individuals formed the release group [81]. After 10 months of acclimatization and pre-training sessions, 31 parrots were released into nature. There was no difference in the survival rate of bolder and shyer individuals, but shyer individuals survived 40 days longer than bolder individuals [81].

The effects of personality traits (boldness, exploration, and sociability) on the survival of European mink (*Mustela lutreola*) after reintroduction were evaluated [49]. The results showed that survival was linked to boldness and exploration, but not to sociability, with bolder animals surviving more than shyer animals, and less exploratory animals surviving more in 2012 and more exploratory animals surviving more in 2013. Initially, cautious animals were favored and then, when they had some knowledge about the environment, bolder animals were apparently favored. Could these results be reflecting a mixed evolutionary stable strategy? [82].

In an evolutionary stable strategy, variation is maintained due to frequency dependent selection, where no rare mutant strategy is favored [83]. Imagine a population of a given animal species, where bolder and shyer individuals can be observed. If the bolder individuals are more aggressive than the shyer individuals, they will win any dispute between them, reproducing more than the shyer ones. Thus, if it is advantageous to be bold, the bold personality will be selected, and bolder individuals will increase in number in that population. However, disputes between bolder individuals may generate physical injuries due to the hyper-aggressiveness shown by both participants in disputes, which would decrease reproductive activities and fitness [84]. In this situation a shyer strategy would be better for individuals, and shyer individuals would increase in that population. Thus, the evolution of different ratios in animal personality types could be explained by game theory [82]. Thus, frequency-dependent selection could explain why in some species shyer individuals do better sometimes and bolder individuals do better in others. However, frequency-dependent selection affects only one aspect that influences personality trait evolution, and other theories are worthy of investigation (for other theories, see [80,82,85]).

Should we release only shyer individuals or only bolder individuals, then? Or should we release a mix of shyer and bolder individuals? Animal personality has a genetic component [86,87,88] and selecting only shyer or only bolder individuals would influence the overall behavioral diversity in a released population [84,85]. Personality diversity is as important as genetic diversity for the establishment and survival of the released population [85,89]. A population with only shyer individuals would result in less reproduction, lower exploration of the environment, and less behavioral diversity [90]. Thus, introducing a mix of shyer and bolder individuals could help in the maintenance of a population subjected to natural and sexual selection [85,89]. As stated earlier, the introduction of a mix of personality profiles would, in theory, facilitate the establishment of a new population, since the best strategy would vary across time, favoring different personality profiles depending on the environmental conditions. The introduction of a mix of personality profiles would work for both social and asocial species.

The boldness dimension has proven to be an interesting starting-point for conservationists to investigate. Maybe a captive experiment could be conducted mixing bold and shy individuals, in different proportions (100% bold; 80% bold and 20% shy; 50% bold and 50% shy; 20% bold and 80% shy; 100% shy) and in different environmental conditions (with predators and without predators) to evaluate how different frequencies of personality traits can affect survival. These examples also show that the study of personality needs to be species-specific, since the results of survival for one species do not necessarily correspond to the factors influencing the survival of a different species. This was a conclusion from an extensive meta-analytical study, which showed that bolder individuals survive more after reintroduction than shyer individuals [91]. However, in terms of species survival in relation to bold or shy individuals surviving better we do not have enough data on different species to analyze if there are other underlying biological causes, such as diet, activity cycle, body size, etc.

### 2.2. Sociability

For social species, the number of social conflicts and how individuals deal with these conflicts can help in the evolution and persistence of personality types. This hypothesis is known as the social niche specialization hypothesis [92]. Thus, it is expected that social species would present more personality types than asocial species, because different personalities would make group living easier [92,93]. A greater number of personality types were found in social shrews [93], spiders [94], and fishes [95,96]. Also, the presence of conspecifics in social species can modify an individual’s willingness to take risks. For the three-spined stickleback fish (*Gasterosteus aculeatus*), the absence of other individuals made solitary animals take more risks [96]. This could be a result of the solitary individual seeking the presence of conspecifics [97,98], the need to explore the environment to locate shelter and food [99], and/or caused by the lack of social information transmission [100,101]. Thus, the release of solitary individuals of social species can increase the risks of failure in a conservation program.

Studies evaluating the frequency of personality traits in nature, the fitness outcomes of each personality trait (i.e., the number of offspring, survival rates, sociability, etc.) and heritability estimates of personality traits [personality traits are known to be heritable, but heritability estimates vary depending on the trait and the species evaluated [88,102]] are important for a better comprehension of the evolution of animal personality. Besides this, some studies show that behavioral syndromes (personality traits at a population level) can vary depending on the habitat type (example: rural or urban) and that personality influences neophilia and neophobia in the individuals (for example urban individuals tend to be bolder, more exploratory and neophiliac, while rural individuals tend to be shyer, less exploratory and neophobic) [103,104]. Such studies could enhance the benefits of using personality as a tool to increase the survival of animals after release.

## 3. Other Links Between Personality and Species Conservation

Personality–conservation studies are not only investigations of the boldness dimension and its link to survival rates after translocation. There are many more aspects of personality that can be studied to enhance conservation success.

### 3.1. Birth Environment

First, the birth environment can shape personality traits in animals. Some studies are linking the birth environment to personality traits of offspring [105]. In these studies, parental effects on their offspring’s phenotypes are discussed. These effects can be both genetical and environmental [105,106]. The parental effect on the offspring’s genotype comes from heritability of genes [88,107,108]. The parental effect on the phenotype of the offspring comes from the influence of the environment on the parents (epigenetic effect), which transmits their responses to the environmental stimuli to their offspring [88,107,108]. Some examples are rank-related facilitation in social species, where the status of the parents in the society can facilitate access to food items, adequate habitats, social raising, and inter-relationships [92]; food provisioning: when parents are able to provide an adequate food supply for the growth of their offspring, this will allow an adequate development of physical and physiological systems, which will influence the expression of certain behaviors later in life [109]; even the place chosen by parents to give birth can influence the personality of the animals, by stimulating the secretion of hormones, that will influence development and behavior [110]. Thus, an animal can be more curious or bolder depending, in part, on parental influence on the birth environment.

### 3.2. Stress

Second, chronic stress can decrease the success of conservation programs and can be linked to personality traits of animals. Distress needs to be avoided for better outcomes in conservation studies since distressed animals have difficulties in adapting to their new habitat, and, therefore, are more prone to be predated [110,111,112,113,114]. However, animals need some acute stress events to enhance physiological responses of their body [79,86] and for the normal elicitation and development of behaviors, such as learning about predators and how to escape them [44,45] or prey-capture behaviors [115]. Some studies link the production of stress hormones and a weak feedback response to personality traits, such as those on the shy-bold continuum axis [example: shyer hens (personality trait) present a higher Hypothalamic-Pituitary-Adrenal activity (hormonal response) and a low rate of feather pecking than bolder hens (behavioral response) [111,116]. This relationship was observed in a study of the great tit (*Parus major*), where more risk-averse individuals were those who produced more glucocorticoid hormones and had a weaker feedback response [111]. Other studies have shown that more risk-averse animals are more prone to develop heart and intestinal diseases, as well as to have a weaker immunological system [112]. For example, optimally stressed animals could better adjust their behavioral and physiological responses to the new environment after release into nature.

### 3.3. Dispersion

Third, could the use of bolder animals to reinforce a population be a better strategy since they disperse and explore more of the environment? This answer depends on other indicators, such as the profile of the natural population, the need to preserve personality diversity, and habitat characteristics, such as predation risk. Bolder and shyer individuals, for example, can present a non-random distribution through the environment, with bolder individuals occupying riskier areas (like those with high human disturbances such as urban environments) and shyer avoiding such areas (personality-matching habitat choice hypothesis [117,118]). Thus, the evaluation of the predation risk of a certain habitat could help to decide which personality profile would be expected to do better after release.

### 3.4. Reproduction

Finally, reproduction can be influenced by personality traits. Some studies link personality characteristics to reproductive success [119,120,121]. Antisocial captive cheetahs (*Acyonix jubatus*), that avoid social interactions reproduce more than social individuals, which was related to ovarian suppression in paired females [122]. In the zebrafish (*Danio rerio*) bolder and more aggressive males fertilize more eggs when compared to shyer individuals [123]. In the giant panda (*Ailuropoda melanoleuca*) more aggressive males produce more cubs due to more successful intromissions in less aggressive females, showing a male/female personality effect on reproduction [38]. The captive breeding of zoo animals using studbooks is based on preserving as much genetic diversity as possible by maintaining the animals under neutral selection conditions [124]. However, in practice this has not always been the case and may not always be the case today. For example, a study found that tamer small cat species were more likely to breed in zoos [125]. Historically, the captive breeding of animals has tried to eliminate individuals who were considered overly aggressive [126], this was actively done for domesticated animals and informally done for zoo animals before the use of studbooks. The insertion of personality data in studbooks could be beneficial in the reproductive management of captive populations [41]. Thus, personality traits can decisively affect reproductive rates and should be considered in animal management [127] (Figure 1).

## 4. How to Evaluate Personality before Release

Since personality is so important for various biological processes, it is important to know the best methods to collect personality data: questionnaires (rating) and behavioral data (coding) [128,129].

The rating method is a qualitative way to consider animal personality [130]. It involves people familiar with the individuals answering questionnaires [131]. In the questionnaires different aspects of animal personality are addressed (boldness, sociability, activity, exploration, aggressiveness), and the respondents score these aspects by marking on scales that normally range from 1 to 10, where 1 means that the aspect is not or hardly observed in the evaluated individual and 10 means that the aspect is always observed in the individual [131,132]. Ratings need to be statistically validated, which is a complicated process [131], because they need to measure animal personality accurately [130]. The rating method is one of the most used in animal personality studies [132]. A recently developed similar approach, which can provide insights into an animal’s personality, is Qualitative Behavioral Assessment [133,134].

The coding method is based on recording the behavior expressed by the individuals during behavioral tests (e.g., open-field, novel object, etc.) or during normal activities, in captivity or in the wild [132]. Behavioral tests are considered better for the animals to display their personality than observing general behavior [128]. Then, the frequencies of behavioral expressions are computed and used in personality determination [129].

Both methods are efficient in personality investigation, and the best method should be chosen based on the study goals and on the environment where the study is being conducted (the rating method is easier for captive environments and the coding method is better for natural environments [128]). The rating method has the advantage of gathering data faster than the coding method, but also has the disadvantage of being based on the judgements of the raters (if more than one rater is rating the animals, their ratings need to present high levels of agreement to be validated; this normally occurs when very experienced raters are rating the animals) [128]. The coding method has the advantage of being based on the behavior exhibited by the animals, which are recorded without the idea of personality classification, but has the disadvantage of being more time-consuming to gather data [128]. Some authors compared the two methods and found contradictory results, with studies showing that both methods generated similar results [135,136,137], and different results [128,138]. Some authors argued that personality researchers should apply both methods because they are complementary and generate important information about the personality dimensions of the animals [139]. It is important to validate behavioral rating and coding because behavioral recordings always involve subjective judgements. Before validity, we need to test the consistency of data, i.e., how reliable is the collected data. Intra-observer, inter-observer and inter-test’s reliability needs to be evaluated by statistical tests, such as correlation statistics, and if the result of the correlation coefficient is 0.7 or more, then we can assume that the data is reliable [140]. Suggestions for data reliability and validation can be found in the scientific literature [130,141].

A practical and quick idea/suggestion for conservationists is to calculate boldness scores for the animals intended to be reintroduced. Boldness score is calculated based on the number of bold and shy behaviors exhibited during novel object trials [67,142]. To do this, researchers need to classify the species’ behaviors into four categories: overly bold, bold, shy, and overly shy. Overly bold behaviors will be summed and multiplied by 3, bold behaviors will be summed and multiplied by 2, shy behaviors will be summed and multiplied by 1 and overly shy behaviors will be summed and multiplied by -1 [142]. The scores for each bold/shy category are summed, and the highest values represent the bolder individuals while the lowest values represent the shyer individuals. This method developed for the swift fox proved to be easy to apply and should be considered in conservation programs (this is only a suggestion; other practices can be found in the scientific literature [127,143]). Obviously, it would be useful to have population data because of the possibility of having only one personality type in your sample.

## 5. Conclusions

Animal personality studies are being applied to animal conservation studies. The evaluation of the influence of birth places, parent personality, stress resilience, and risk assessment can be important to select the best individuals to be released. Behavioral tests, such the novel object test, should be implemented to discriminate personality aspects of the individuals, such as boldness and aggressiveness. Personality studies could help conservation programs to increase their success rate.

## Figures and Tables

**Figure 1 animals-11-01019-f001:**
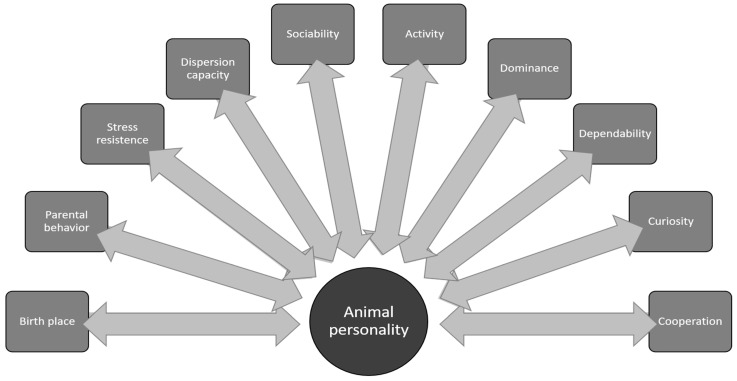
Flowchart representing the different factors that influence and are influenced by animal personality.

## Data Availability

No dataset was created.

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
