# Peer review of "Animal Personality and Conservation: Basics for Inspiring New Research"

_animals, 2021, doi:10.3390/ani11041019_

Round 1

Reviewer 1 Report

The authors have reviewed a growing area of research that is of great potential utility for conservation science. However, there are some substantial issues with the organization of the paper. The introduction mainly covers translocation programs, but that is only one of the conservation activities that can be informed by personality research (and in fact is only the first section of the review, as section 3 is meant to cover these other topics).  The introduction sort of simplifies the study of personality; for example, by identifying particular dimensions as if it is an exhaustive list. Section 2 covers some interesting studies, but the organization of this section is unclear, sometimes just switching from one paragraph describing one study to a new paragraph describing another study without an overarching topic. To make a more useful contribution to the literature, it would help if this section were divided in some logical way, either by specific personality dimensions or by particular hypotheses that can explain how personality interacts with success at release. Something more organized will help generate meaningful hypotheses to test, which seems to be the aim of this review paper. Consider adding subheadings to make these divisions clearer to the reader. The same comments apply to section 3. Section 4 is so general that it is not particularly useful. In general, the paper needs to be heavily rewritten, both for English language grammar and for organizational purposes. Right now, the review is not structured in such a way that the reader can easily glean new hypotheses and approaches to studying the effect of personality on conservation, so it may not make a meaningful addition to the scientific literature in this area.  

Line 16: “the ambient” ...environment? There seems to be a word missing here.

Line 27-28: The wording of this sentence is confusing, particularly what is meant by “mapping” personality characteristics of individuals.

Line 81: should be “coping style”

Lines 126-132: This paragraph conflates boldness and exploration; although in truth, it can be difficult to differentiate between them. This type of nuance could use further discussion when considering the results that are summarized here in relation to personality and translocation success.

Lines 248-256: This section revisits the issue of translocation and therefore seems more appropriate in section 2.

Line: 260-261: Please rethink this reference and the context it was presented in. For starters, “African” is not really a useful cultural designation, considering the diversity of nations, cultures, and languages represented by this region. Secondly, it comes off as very imperialist/ethnocentric to group polygynous Africans into this list of references, the remainder of which all deal with nonhuman animals. It calls back old tropes that current African cultures represent something pre-human compared to western civilization.  It is uncomfortable at best and at worst, comes across as racist.

Lines 276-279: Talking about personality in relation to captive breeding is slightly outside the bounds of this review, in my opinion, given that most captive-bred animals will never be reintroduced to the wild.

Author Response

The authors have reviewed a growing area of research that is of great potential utility for conservation science. However, there are some substantial issues with the organization of the paper. The introduction mainly covers translocation programs, but that is only one of the conservation activities that can be informed by personality research (and in fact is only the first section of the review, as section 3 is meant to cover these other topics).

Response: we are using the term animal translocations for grouping all the other conservation activities (see lines 38-43).

The introduction sort of simplifies the study of personality; for example, by identifying particular dimensions as if it is an exhaustive list.

Response: We inserted the other aspects of animal personality in this section.

Section 2 covers some interesting studies, but the organization of this section is unclear, sometimes just switching from one paragraph describing one study to a new paragraph describing another study without an overarching topic. To make a more useful contribution to the literature, it would help if this section were divided in some logical way, either by specific personality dimensions or by particular hypotheses that can explain how personality interacts with success at release. Something more organized will help generate meaningful hypotheses to test, which seems to be the aim of this review paper. Consider adding subheadings to make these divisions clearer to the reader. The same comments apply to section 3.

Response: we inserted subheadings to make the topics clearer.

Section 4 is so general that it is not particularly useful.

Response: we do not agree because this section is about how to measure animal personality. Since neither the editor nor the other 2 reviewers suggested to delete section 4, we decided to maintain it in the text.

In general, the paper needs to be heavily rewritten, both for English language grammar and for organizational purposes. Right now, the review is not structured in such a way that the reader can easily glean new hypotheses and approaches to studying the effect of personality on conservation, so it may not make a meaningful addition to the scientific literature in this area.

Response: the paper was English revised by one of the authors (a Native English speaker).  

Line 16: “the ambient” ...environment? There seems to be a word missing here.

Response: changed.

Line 27-28: The wording of this sentence is confusing, particularly what is meant by “mapping” personality characteristics of individuals.

Response: we changed this sentence to clarify.

Line 81: should be “coping style”

Response: changed.

Lines 126-132: This paragraph conflates boldness and exploration; although in truth, it can be difficult to differentiate between them. This type of nuance could use further discussion when considering the results that are summarized here in relation to personality and translocation success.

Response:

Lines 248-256: This section revisits the issue of translocation and therefore seems more appropriate in section 2.

Response: with the adjustments made in this section, we believe that now this paragraph is suitable for this section.

Line: 260-261: Please rethink this reference and the context it was presented in. For starters, “African” is not really a useful cultural designation, considering the diversity of nations, cultures, and languages represented by this region. Secondly, it comes off as very imperialist/ethnocentric to group polygynous Africans into this list of references, the remainder of which all deal with nonhuman animals. It calls back old tropes that current African cultures represent something pre-human compared to western civilization.  It is uncomfortable at best and at worst, comes across as racist.

Response: authors agreed in deleting this part.

Lines 276-279: Talking about personality in relation to captive breeding is slightly outside the bounds of this review, in my opinion, given that most captive-bred animals will never be reintroduced to the wild.

Response: we do not agree with this statement. So, we decided to keep this part in the text. Captive animals are considered resources for animal conservation programs and are part, although not always, of conservation programs (Brichieri-Colombi et al., 2018. Conservation Biology33(1): 33-39), and personality management is a reality that can help zoos in their conservation planning.

Reviewer 2 Report

This paper reviews the role of personality in the success of translocation of animals as a conservation tool.  It provides a very useful summary of the current state of the art of the use of personality in recent translocations as well as a discussion of the factors that influence personality and determine which kinds of personality affect translocation success.  A valuable learning point form this review is that the impact of personality on translocation success is species specific and may vary between areas and between years depending on environmental conditions and the presence/personalities of conspecifics.  The authors recommend methodologies for assessing personality and evaluate their advantages and disadvantages.  This paper is a valuable review for conservation biologists involved in translocations. 

Overall I have no major criticisms of this paper. but I do have one comment:

On lines 236-237, it would be worth also noting that stressful experiences may be needed for the normal elicitation and development of behaviours, such as learning about predators and how to escape them, or how to deal with difficult prey, which may need to occur at critical times during normal behavioural development. 

I have made numerous minor comments on the manuscript attached.

Author Response

This paper reviews the role of personality in the success of translocation of animals as a conservation tool.  It provides a very useful summary of the current state of the art of the use of personality in recent translocations as well as a discussion of the factors that influence personality and determine which kinds of personality affect translocation success.  A valuable learning point form this review is that the impact of personality on translocation success is species specific and may vary between areas and between years depending on environmental conditions and the presence/personalities of conspecifics.  The authors recommend methodologies for assessing personality and evaluate their advantages and disadvantages.  This paper is a valuable review for conservation biologists involved in translocations. 

Overall I have no major criticisms of this paper. but I do have one comment:

On lines 236-237, it would be worth also noting that stressful experiences may be needed for the normal elicitation and development of behaviours, such as learning about predators and how to escape them, or how to deal with difficult prey, which may need to occur at critical times during normal behavioural development. 

Response: inserted.

I have made numerous minor comments on the manuscript attached.

Response: all suggestions were incorporated in this new version.

Reviewer 3 Report

This paper is a review of the literature relating personality in animals to success in conservation efforts, particularly translocation and reintroduction efforts. The main premise is that knowledge of an animal’s personality might lead to greater success. In many ways, this is a topic whose time has come, and the authors do a reasonable job of summarizing the small literature that there is on the topic (certainly, the examples they give are compelling, but I am not an expert in the translocation/reintroduction area, so cannot speak to whether the review is comprehensive). But the review is uneven. There is some good, sophisticated, and nuanced discussion about how translocation efforts might incorporate such information, but there are a couple of places where the treatment seems too superficial. I will focus on two:

The first issue concerns the listing of personality dimensions on lines 85-87. First, I don’t think any personality researcher would say that “Personality is divided into dimensions...” This kind of phrasing suggests that personality is a “thing” in the brain of the animal, and it can be divided up. Nothing could be further from the truth. Personality is a hypothetical construct that is purely descriptive. It is used to describe dispositions or motivations that animals can have and that can differ between animals. Second, the authors label animal personality dimensions as those derived from the Big Five model of human personality. But not all animal personality researchers have adopted that framework, for a variety of reasons. In fact, there are long-standing research programs that have identified personality factors with different names; admittedly, many dimensions overlap with those named by the current authors. The authors should be somewhat more inclusive by not stating so forcefully that these are *the* dimensions for animal personality. And it might be useful to let the reader know that there are other named dimensions that do overlap conceptually with those listed. For example, there is quite a bit of primate work done with the concept “sociability.” That is not listed in the authors’ list, but it is similar to extraversion/agreeableness. The disconnect with the naming of animal personality dimensions is evident throughout the rest of the review, which focuses primarily on “bold,” “shy,” and “exploratory” dimensions none of which are mentioned in the authors’ description here of the personality dimensions. In fact, in the section on measuring personality, there is a listing of animal personality dimensions (lines 288-289) and none of those listed (except for activity) overlap with the listing on pp. 85-87. So overall, I think the authors should spend more time here discussing some of these nuances, for those readers who are not already deep into the animal personality literature.

The second issue involves the discussion of measuring personality, which appears in section 4. The discussion here seems superficial. Yes, there are two broad ways of quantifying personality, using ratings or coding of specific behaviors. But the discussion of the coding approach focuses on behavioral *tests*. While behavioral tests are often used, they are by no means the only way that behavioral coding has been used to identify personality dimensions – in many studies, animals’ natural behavior, in its natural setting, is observed (and perhaps coded) before ratings are made – no behavioral tests involved. In addition, in lines 309-311, the authors write “The coding method has the advantage to not be based on researchers’ judgements, because the behavior exhibited by the animals are recorded without the idea of personality dimensions...” I understand what the authors are trying to say here, but I don’t believe this is the best way to say it. “Judgement” is involved in any kind of behavioral data collection, whether it is with ratings or with coding of behavior. For the latter, judgements include whether a given behavior meets a definition in the observer’s ethogram. If there were no judgements involved, there wouldn’t be a need to generate inter-observer reliability data, which is all about insuring that two people make the SAME judgements. This sentence should be rephrased. But more generally, the authors miss an opportunity to make the reader aware of issues of reliability and validity, which bear on BOTH methods of personality assessment. So this section could be fleshed out better.

More minor points:

l. 47: sentence structure: Conservation biologists were scientifically separated from behavioral biologists after long...

L. 81: “copying styles” should be coping styles?

L. 87: sentence structure: “Although relatively new for science...” It’s unclear what this phrase is referring to, the newness of the concept of animal personality or the newness of linking personality to conservation.

L. 133: sentence structure: “influence of the personality on the survival” should be “influence of personality on the survival”...

L. 151: sentence structure: “Thus, it is advantageous to be bold...” should be, I believe, “Thus, if it is advantageous to be bold...”

L. 157: “difference” should be “different.”

L. 224: “social facilitation” is mentioned and described as a process whereby the status of the parents can facilitate access to resources. Suggest the authors use a different term for what they are trying to get at here, since “social facilitation” is a long-standing concept in Social Psychology that bears no relationship to the present authors’ use of the term – specifically, it refers to an improvement in the effort/performance of an individual when in the presence of others, versus when alone. This is quite different from anything rank-related.

L. 255-256: “Thus, the evaluation of the habitat risks could help for decide of what personality profile would be expected to do better after release.”

L. 258 ff: “reproduction success,” I think the authors mean “reproductive success.”

Author Response

This paper is a review of the literature relating personality in animals to success in conservation efforts, particularly translocation and reintroduction efforts. The main premise is that knowledge of an animal’s personality might lead to greater success. In many ways, this is a topic whose time has come, and the authors do a reasonable job of summarizing the small literature that there is on the topic (certainly, the examples they give are compelling, but I am not an expert in the translocation/reintroduction area, so cannot speak to whether the review is comprehensive). But the review is uneven. There is some good, sophisticated, and nuanced discussion about how translocation efforts might incorporate such information, but there are a couple of places where the treatment seems too superficial. I will focus on two:

The first issue concerns the listing of personality dimensions on lines 85-87. First, I don’t think any personality researcher would say that “Personality is divided into dimensions...” This kind of phrasing suggests that personality is a “thing” in the brain of the animal, and it can be divided up. Nothing could be further from the truth. Personality is a hypothetical construct that is purely descriptive. It is used to describe dispositions or motivations that animals can have and that can differ between animals.

Response: we changed the idea of the division of personality into dimensions. We inserted that personality is a construct, identified and classified into: …..  See lines 85-88.

Second, the authors label animal personality dimensions as those derived from the Big Five model of human personality. But not all animal personality researchers have adopted that framework, for a variety of reasons. In fact, there are long-standing research programs that have identified personality factors with different names; admittedly, many dimensions overlap with those named by the current authors. The authors should be somewhat more inclusive by not stating so forcefully that these are *the* dimensions for animal personality. And it might be useful to let the reader know that there are other named dimensions that do overlap conceptually with those listed. For example, there is quite a bit of primate work done with the concept “sociability.” That is not listed in the authors’ list, but it is similar to extraversion/agreeableness. The disconnect with the naming of animal personality dimensions is evident throughout the rest of the review, which focuses primarily on “bold,” “shy,” and “exploratory” dimensions none of which are mentioned in the authors’ description here of the personality dimensions. In fact, in the section on measuring personality, there is a listing of animal personality dimensions (lines 288-289) and none of those listed (except for activity) overlap with the listing on pp. 85-87. So overall, I think the authors should spend more time here discussing some of these nuances, for those readers who are not already deep into the animal personality literature.

Response: we inserted sentences citing the other dimensions identified for animals.

The second issue involves the discussion of measuring personality, which appears in section 4. The discussion here seems superficial. Yes, there are two broad ways of quantifying personality, using ratings or coding of specific behaviors. But the discussion of the coding approach focuses on behavioral *tests*. While behavioral tests are often used, they are by no means the only way that behavioral coding has been used to identify personality dimensions – in many studies, animals’ natural behavior, in its natural setting, is observed (and perhaps coded) before ratings are made – no behavioral tests involved.

Response: we inserted this in the text.

In addition, in lines 309-311, the authors write “The coding method has the advantage to not be based on researchers’ judgements, because the behavior exhibited by the animals are recorded without the idea of personality dimensions...” I understand what the authors are trying to say here, but I don’t believe this is the best way to say it. “Judgement” is involved in any kind of behavioral data collection, whether it is with ratings or with coding of behavior. For the latter, judgements include whether a given behavior meets a definition in the observer’s ethogram. If there were no judgements involved, there wouldn’t be a need to generate inter-observer reliability data, which is all about insuring that two people make the SAME judgements. This sentence should be rephrased. But more generally, the authors miss an opportunity to make the reader aware of issues of reliability and validity, which bear on BOTH methods of personality assessment. So this section could be fleshed out better.

Response: we inserted information about data reliability and validity in personality studies and changed the part about judgments.

More minor points:

  1. 47: sentence structure: Conservation biologists were scientifically separated from behavioral biologists after long...

Response: we restructured this sentence.

  1. 81: “copying styles” should be coping styles?

Response: changed.

  1. 87: sentence structure: “Although relatively new for science...” It’s unclear what this phrase is referring to, the newness of the concept of animal personality or the newness of linking personality to conservation.

Response: this is referring to the newness of linking animal personality and conservation. We changed this sentence to make it clear.

  1. 133: sentence structure: “influence of the personality on the survival” should be “influence of personality on the survival”...

Response: changed.

  1. 151: sentence structure: “Thus, it is advantageous to be bold...” should be, I believe, “Thus, if it is advantageous to be bold...”

Response: changed.

  1. 157: “difference” should be “different.”

Response: changed.

  1. 224: “social facilitation” is mentioned and described as a process whereby the status of the parents can facilitate access to resources. Suggest the authors use a different term for what they are trying to get at here, since “social facilitation” is a long-standing concept in Social Psychology that bears no relationship to the present authors’ use of the term – specifically, it refers to an improvement in the effort/performance of an individual when in the presence of others, versus when alone. This is quite different from anything rank-related.

Response: we changed for rank-related facilitation.

  1. 255-256: “Thus, the evaluation of the habitat risks could help for decide of what personality profile would be expected to do better after release.”

Response: we changed this sentence to clarify.

  1. 258 ff: “reproduction success,” I think the authors mean “reproductive success.”

Response: changed.

Round 2

Reviewer 3 Report

This is a greatly improved version of the manuscript. There is still superficial treatment of some topics (eg, reliability and validity), but references are provided for readers that want more information.  I think this will be a watershed paper in this area.